# Single-Cell Sequencing Confirms Transcripts and V_H_DJ_H_ Rearrangements of Immunoglobulin Genes in Human Podocytes

**DOI:** 10.3390/genes12040472

**Published:** 2021-03-25

**Authors:** Zhenling Deng, Huige Yan, Zhan Shi, Xinyu Tian, Zhuan Cui, Yingchun Sun, Song Wang, Danxia Zheng, Xiaoyan Qiu, Yue Wang

**Affiliations:** 1Department of Nephrology, Peking University Third Hospital, Beijing 100191, China; dengzhenling1985@bjmu.edu.cn (Z.D.); xytian24@163.com (X.T.); drcuizhuan@163.com (Z.C.); 15245094539@163.com (Y.S.); songwang30@163.com (S.W.); m15611908597@163.com (D.Z.); 2Department of Immunology, School of Basic Medical Sciences, Peking University, Beijing 100191, China; 1310305230@pku.edu.cn (H.Y.); 1310305224@pku.edu.cn (Z.S.)

**Keywords:** podocyte, single-cell RNA sequencing, immunoglobulins, gene transcription, V_H_DJ_H_ rearrangement

## Abstract

Most glomerular diseases are associated with inflammation caused by deposited pathogenic immunoglobulins (Igs), which are believed to be produced by B cells. However, our previous study indicated that the human podocyte cell line can produce IgG. In this study, we aimed to confirm the transcripts and characterize the repertoires of Igs in primary podocytes at single cell level. First, single-cell RNA sequencing of cell suspensions from “normal” kidney cortexes by a 10xGenomics Chromium system detected Ig transcripts in 7/360 podocytes and Ig gene segments in 106/360 podocytes. Then, we combined nested PCR with Sanger sequencing to detect the transcripts and characterize the repertoires of Igs in 48 single podocytes and found that five classes of Ig heavy chains were amplified in podocytes. Four-hundred and twenty-nine V_H_DJ_H_ rearrangement sequences were analyzed; podocyte-derived Igs exhibited classic V_H_DJ_H_ rearrangements with nucleotide additions and somatic hypermutations, biased VH1 usage and restricted diversity. Moreover, compared with the podocytes from healthy control that usually expressed one class of Ig and one V_H_DJ_H_ pattern, podocytes from patients expressed more classes of Ig, V_H_DJ_H_ patterns and somatic hypermutations. These findings suggested that podocytes can express Igs in normal condition and increase diversity in pathological situations.

## 1. Introduction

Podocytes play important roles not only in the glomerular filtration barrier but also in both innate and adaptive immunity. By expressing complement and related receptors, toll-like receptors, major histocompatibility complex I/II and co-stimulatory molecules such as CD80, podocytes participate in the renal immune response [1]. 

It was once generally thought that immunoglobulins (Igs) are produced solely by mature B cells and plasma cells and acted as antibodies to recognize and neutralize various pathogens. However, this theory has been challenged over the past decades by accumulating evidence reporting that Igs could be expressed in non-B cells, including a variety of tumor cells [2,3,4] and normal cells, even in so-called immune privileged sites including the eyes [5], central neurons [6,7], placenta [8], testes [9] and mammary epithelial cells during lactation [10]. In contrast to B-Igs, non-B-Igs present limited diversity [11] and physicochemical properties, such as abnormal glycation [12] and hydrophobic properties. Functionally, the non-B-Igs can serve as natural antibodies in skin and mucosa. For example, skin epidermal cell-derived IgG and IgA showed natural antibody activity by binding pathogens such as Staphylococcus aureus [13]; Epithelial cell-derived IgM showed polyreactivity, which could bind ssDNA, dsDNA, LPS, insulin and different types of microbes [14]. In addition, non-B-Igs can serve as growth factors to promote cell proliferation-and adhesion, and the migration, invasiveness and metastasis of cancer cells [15], thereby suggesting that non-B Igs participate in tumorigenesis and development.

Our previous studies have demonstrated that human renal mesangial cells [16], podocytes [17] and proximal tubular epithelial cells [18] can produce IgA or IgG, which are involved in cell growth/adhesion and are up-regulated by angiotensin II, *Staphylococcus aureus* and TGF-β1. Given that these immortalized cells may lose their original characteristics in the human kidney, in this study, we aimed to confirm the transcripts and repertoire of Igs in primary single podocytes. The 10× Genomics Chromium system and nested PCR combined with Sanger sequencing were utilized. Five classes of Ig heavy chains were amplified in single podocytes. In addition, single podocytes from the patients with kidney diseases exhibited more classes of Igs and more V_H_DJ_H_ patterns, along with higher somatic hypermutation.

## 2. Materials and Methods

### 2.1. Patients and Control Subjects

This study conformed to the principles of the Helsinki declaration, and was approved by the Medical Ethics Committee of Peking University Third Hospital and conducted in accordance with the protocol. All donors voluntarily donated kidney cortexes and signed the informed consent forms prior to donating the kidney cortexes to the study. All methods were carried out in accordance with relevant guidelines and regulations. These samples were strictly anonymized.

Kidney cortexes from 2 patients undergoing nephrectomy as a result of renal or ureteral carcinoma, 4 patients with IgA nephropathy (IgAN), 3 patients with membranous nephropathy (MN) and 1 patient with ischemic nephropathy were prepared for single-cell RNA sequencing (scRNA-seq). All patients had new-onset diseases and had not been treated with glucocorticoid, immunosuppressor or antineoplastic drugs. Patients with a history of hepatitis B/C, diabetes, rheumatic immune diseases or infection within the three months before biopsy were excluded. Clinical data submitted at the time of nephrectomy or renal biopsy are summarized in the Appendix A. 

### 2.2. 10× Library Preparation and Sequencing 

Considering the abundance of tubular epithelial cells and the small amount of glomerular intrinsic cells in a normal kidney cortex, we enriched glomeruli by sequential filtration through 80 and 140 mesh sieve, prepared single-cell suspensions by digesting glomeruli with collagenase I and subsequently performed scRNA-seq and V(D)J-seq using the Single-Cell Immune Profiling Solution. The concentration of the single-cell suspension was counted and adjusted to 1000 cells/μL for a capture of 7000 cells. All remaining procedures, including library construction, were performed according to the manufacturer’s standard protocol described in Shi’ work [19]. We used the Cell Ranger software pipeline (version 3.0.0, 10xGenomics, USA) to demultiplex cellular barcodes and map reads to the genome. Loupe Browser (version 4.0.0, 10xGenomics, USA) was used for clustering. The barcodes of podocytes were obtained by the clustering of the 10× Genomics transcriptome, and the fastX-Toolkit (Version 0.0.13, Cold Spring Harbor Laboratory, USA) was used to split the data of immune repertoire and obtained Ig sequences expressed in podocytes. Then MiXCR (Version 3.0.7, MiLaboratory, Russia) was used for mapping analyses of Ig genes with accurate alignment of gene segments.

### 2.3. Tissue Processing and Single-Cell Dissociation 

The biopsy specimen was finely minced into small pieces. Single glomeruli were isolated under a microscope by capillary pipette and digested in PBS containing 1 mg/mL collagenase I (Sigma Chemicals, St. Louis, MO, USA) for 5 min at 37 °C. Glomerular intrinsic cells were released and a single cell was manually picked out under the microscope by capillary pipette. 

### 2.4. RNA Extraction and cDNA Synthesis of Single Cells

Glomerular single-cell RNA extraction and cDNA synthesis were carried out by Tang’s methods as described [20]. Briefly, glomerular single cells were isolated and put into lysate buffer. Reverse transcription was then performed directly on the whole cell lysate. After this, the free primers were removed by ExoSAP-IT and a poly(A) tail was added to the 3′ end of the first-strand cDNA by Terminal Deoxynucleotidyl Transferase. Then the single- cell cDNAs were amplified with the following PCR program: 95 °C for 3 min; then 25 cycles of 95 °C for 30 s, 67 °C for 1 min, and 72 °C for 6 min (+6 s each cycle). After this step, all cDNAs had been amplified. 

### 2.5. PCR Amplification

The variable regions of IgH were amplified by a nested PCR. Concretely, the complete V_H_DJ_H_ rearrangement of IgH were amplified from single-cell cDNA with an upstream variable-region primer pool for VH1-FR1, VH2-FR1, VH3-FR1, VH4-FR1, VH5-FR1 and VH6-FR1 and downstream constant-region primer for IGHG-R1, IGHA-R1, IGHM-R1, IGHD-R1 and IGHE-R1, to amplify Ig γ, Ig α, Ig µ, Ig δ and Ig ε, respectively. For the second-round PCR, an upstream primer that anneals to the framework 2 (FR2) region coupled with a JH primer was used to amplify the variable region of IgH. Touchdown PCR was performed for single-cell IgH amplification. Nested PCR was also performed to detect podocyte specific marker genes (NPHS1and NPHS2), and B cell marker gene (CD19). Primers and PCR conditions of IgG4 were as described in Jing [17]. The PCR products were separated on a 2% agarose gel by electrophoresis. The primer pool of upstream variable-region refers to primers in Biomed-2 [21]. Other primers used for PCR are listed in the Appendix A.

### 2.6. Sanger Sequencing and Analyses of the Sequencing Data 

PCR products were cloned into a pGEM-T Easy Vector (Promega, Madison, WI) and transformed into DH5a-competent bacteria (Invitrogen). In all, 5–15 colonies per sample were chosen randomly, and sequenced with an ABI 3100 Genetic Analyzer (Applied Biosystems, Foster City, CA, USA). The rearranged V_H_DJ_H_ sequences were compared with those in the basic local alignment search tool (IgBLAST, https://www.ncbi.nlm.nih.gov/igblast/, accessed on 24 March 2021) to identify the best matching germline gene segments and junctions following primer trimming. For the analysis of somatic hypermutation, the part of the IgH V region from FR2 to JH was used. Mutation status was designated as mutated if there were 2% or more mutations (<98% homology to germline sequences) compared with the germline sequences. To analyze whether the mechanism of somatic hypermutation (SHM) occurring in the glomerular-derived Ig variable region was similar to that caused by antigen selection in B-Igs, the mutation frequency of the RGYW motifs (R = A or G; Y = C or T; W = T or A) used as a principal hotspot for AID induced G:U lesions was calculated. A hotspot mutation ratio above 25% is considered to be antigen-induced SHM.

### 2.7. Data Mining of Single-Cell RNA Sequencing Datasets

The data of single B cells RNA-seq were retrieved from the single-cell data website of 10× Genomics (https://support.10xgenomics.com/single-cell-vdj/datasets/2.2.0/vdj_v1_hs_pbmc_b, accessed on 24 March 2021) [22]. We screened 793 CD19+B cells, linked the 5′ RNA-seq data to the VDJ data, and obtained 852 VDJ sequences (including 54 unproductive sequences). The frequencies of VH, D, JH usage in single B cells were analyzed and compared with those in podocytes. The single-cell database from a human allograft kidney biopsy undergoing mixed rejection (GSE109564) [23] and one built from isolated glomerular cells from C57BL/6 (GSE111107) [24] were mined.

### 2.8. Statistical Analysis 

The significant differences in somatic hypermutation among groups were calculated by Mann-Whitney U test. Statistical significance was defined as *p* < 0.05. **** *p* < 0.0001, *** *p* < 0.001, ** *p* < 0.01, * *p* < 0.05.

## 3. Results

### 3.1. Few Ig Gene Transcripts and Lots of V, D or J Gene Segments Were Detected in Single Podocytes by the 10×Genomics Chromium System

In order to obtain a broad picture of the transcripts and immune repertoire of Igs in podocytes, we first performed scRNA-seq and V(D)J-seq using a Single-Cell Immune Profiling Solution on the prepared single-cell suspension from the enriched glomeruli of the kidney cortex from a patient undergoing nephrectomy as a result of ureteral carcinoma. Unsupervised clustering analysis of the single cells by Loupe Browser identified 11 distinct cell clusters (Figure 1A). On the basis of specific marker genes NPHS1, NPHS2, WT1, SYNPO and PODXL, 360 single cells were defined as podocytes with low expression of mitochondrial genes. Transcriptome data analysis showed that only seven podocytes expressed Ig genes, including heavy chain and light chains, a constant regions or variable regions, such as IGHG3, IGHV3-7, IGHV3-30, IGHV3-74, IGKC and IGKV1-17 (Figure 1B). V(D)J-seq analyzed the transcripts of V, D and J segments from podocytes and found that 29.4% (106/360) of podocytes expressed Ig genes, including heavy chains (IGHA, IGHG, IGHM and IGHD), light chains (IGKC and IGLC) and some predominant variable region segments (IGHV3-11, IGHD2-2, IGHJ4, IGKV1-17 and IGKJ3) (Figure 1C).

### 3.2. Transcripts of Five Classes of Ig Heavy Chain Genes Were Amplified in Single Podocytes by Nested PCR and SANGER Sequencing

Given that it was difficult to detect low-abundance genes by high-throughput scRNA-seq, we turned to nested PCR combined with Sanger sequencing to explore the Ig transcription in single podocytes. The kidney cortex of a patient undergoing nephrectomy as a result of renal carcinoma was minced into small pieces and the glomeruli were isolated (Figure 2A,B). After digesting the glomeruli with collagenase I, a total of 235 glomerular single cells were manually picked by capillary tube under the microscope, of which 48 single cells were identified as podocytes that expressed Ig gene transcripts. Igα, γ, δ, μ-and ε were all amplified in podocytes (Figure 2C,D) with detection frequencies of 60.4% for IgG, 56.3% for IgM, 12.5% for IgA, 10.4% for IgE and 4.2% for IgD (Figure 2E, Appendix A).

### 3.3. Podocyte-Derived Igs Presented Some Basic Characteristics Similar to Those of B-Igs

To determine the repertoire of podocyte-derived Igs, 429 sequences of V_H_DJ_H_ rearrangements from 48 single podocytes were analyzed (Appendix A). Podocyte-derived Ig heavy chains displayed functional V_H_DJ_H_ rearrangements (functional rate of 95.8%, 411/429, Appendix A) with classic nucleotide additions at the V–D junctions and D–J junctions (Figure 3A), and most single podocytes displayed unique V_H_DJ_H_ rearrangement patterns (Appendix A). Moreover, the somatic hypermutations (SHM) and the hotspot motif mutations (antigen-induced SHM at RGYW/WRCY, W = A/T, R = A/G, and Y = C/T) were detected in IgH variable region genes with rare IgD mutations, a low level of IgM mutation and high levels of mutation in IgA, IgG and IgE (Figure 3B,C). All these features are similar to those of B cells.

### 3.4. Podocyte-Derived Igs Displayed VH1 Bias and Lower Diversity Than B-Igs

Although most single podocytes displayed unique V_H_DJ_H_ rearrangements, four pairs of single podocytes from different individuals shared the same V_H_DJ_H_ patterns with identical junctions (Figure 4A), which is impossible for B cells in such a small number. The length distribution of the complementarity determining region 3 (CDR3), another indicator of Ig diversity, in B cells was close to a Gaussian distribution (*R*^2^ = 0.91), while the CDR3 length in podocytes was far from a Gaussian distribution (*R*^2^ = 0.46, Figure 4B). These suggested that podocyte-derived Igs displayed less diversity than B-Igs.

We compared the frequency of the VH, D- and JH usage in podocytes and B cells and found that the gene fragments and distributions of DH and JH families were overall similar to those in B cells except that the VH1 was used with a higher frequency in podocytes (38.5% in podocytes vs. 12.9% in B cells). The top two VH usages in podocytes were VH3 (39.6%) and VH1 (38.5%), while those in B cells were VH3 (52.3%) and VH4 (25.1%) (Figure 5A–F). The most used VH-gene fragment in podocytes was IGHV1-18 (17.8%), whereas that in B cells was IGHV3-23 (13.5%) (Figure 5G).

### 3.5. Podocytes from Kidney Patients Exhibited More Ig Classes, More V_H_DJ_H_ Patterns and Higher SHM

Despite the small number of single podocytes in each group, the difference between healthy control and kidney patients was significant. First, compared with only one class of IgH in “normal” single podocytes, 40.5% (17/42) co-expression of IgH was detected in patients’ single podocytes. One third of podocytes had co-expression of IgG and IgM in patients with MN and IgAN. Four out of nine single podocytes had co-expression of IgM and IgE, or IgA and IgG, or IgG, IgM and IgE, or even IgA, IgM, IgD and IgE in patients with ischemic nephropathy (Figure 6A,B). In addition, we found that 50% (24/48) of single podocytes can express more than one functional V_H_DJ_H_ pattern: 33.3% (2/6) from the healthy control and 52.4% (22/42) from the kidney patients, including 61.1% (11/18) from MN, 55.6% (5/9) from ischemic nephropathy and 40% (6/15) from IgAN. Moreover, V_H_DJ_H_ sequences of different IgH in a single podocyte did not share the same V_H_DJ_H_ pattern, suggesting that podocyte-derived V_H_DJ_H_ sequences do not originate from the classical class switching (Figure 6C,D). More importantly, we compared the differences of SHM among the different groups and found that the SHM frequency in kidney patients was significantly higher than that in the healthy controls (Figure 6E, *p* < 0.001).

### 3.6. Confirmation of Ig Transcription in Podocytes by Published scRNA-seq Data

Recently, tens of sets of kidney scRNA-seq data have been released for humans and mice. We utilized NPHS1, NPHS2, WT1 and SYNPO as marker genes to identify podocytes from scRNA-seq data. In the single-cell database from an allograft kidney biopsy (GSE109564) [23], we found that IgH such as Ig α, γ, δ and μ and light chains κ and λ were expressed in human podocytes, and there were even multiple full IgH complexes (including Igγ and Igα or Igδ) in a single podocyte (Figure 7A). In the single-cell RNA-seq database of isolated glomerular cells from C57BL/6 (GSE111107) [24], IgH such as Igα, γ and μ and light chains κ and λ were found in single mouse podocytes (Figure 7B).

## 4. Discussion

In this study, we found that all the five classes of Ig heavy chains were amplified in primary single podocytes. Similar to B-Igs, podocyte-derived Igs have shown classical V(D)J rearrangements with nucleotide insertions at junctions and somatic hypermutation. On the other hand, podocyte-derived Igs demonstrate some distinct features, such as biased VH1 usage, skewed distribution of CDR3 lengths and restricted diversity. In addition, single podocytes from the patients with kidney diseases exhibited more classes of Igs and more V_H_DJ_H_ patterns, along with higher somatic hypermutation. 

In order to obtain an overall picture of the transcripts and immune repertoire of Igs in podocytes, we first performed high-throughput scRNA-seq and found the transcripts of Ig genes in only a few podocytes. At the same time, many V, D or J gene segments were detected in podocytes by V(D)J sequencing, but no functional V(D)J rearrangements. The evidence that podocytes express immunoglobulins was not adequate due to the limitation of the 10×Genomics Chromium system in detecting lower abundance genes. To be specific, the IgG gene abundance of non-B cells is relatively low, so the 10× Genomics technology is not sensitive enough to the detection of low abundance genes. Therefore, the evidence for Igs gene transcription was obtained from such a small number of podocytes although high throughput sequencing technologies were utilized in this study and in other researches on scRNA-seq of kidney. Nested PCR amplified by two rounds of PCR increased the sensitivity of detecting low abundance genes. In addition, because of the diversity of immunoglobulin gene rearrangements, it is difficult to reassemble the intact V(D)J sequences which has been disrupted into small segments before sequencing. Sanger sequencing directly sequenced large fragments of Ig genes, without interruption in advance, avoiding the problems caused by Ig gene segmentation and reassemble.

The time-consuming nested PCR and Sanger sequencing brought us a great surprise. We spent a lot of time manually picking single podocytes via capillary pipette and amplifying the transcripts of Igs in single podocytes. All the five classes of Igs were successfully detected in podocytes with the frequency ranking (from high to low) of IgG, IgM, IgA, IgE and IgD. Moreover, the repertoire of Igs in podocytes was also revealed as a result of the sequencing of a large quantity of V(D)J rearrangements. On the one hand, podocyte-derived Igs presented some basic characteristics similar to B-Igs, such as the classical and functional V(D)J rearrangements, somatic hypermutation (SHM) and the hotspot motif mutation. In addition, less diversity of V(D)J rearrangements and biased VH1 usage were also found in podocyte-derived Igs. All these data further confirmed that Ig genes transcription and rearrangements do exist in podocytes. Given the low abundance of Igs genes in podocytes, the dynamic expression in different cell cycles and limited marker genes for podocyte identification, our results may underestimate the frequency of Ig transcripts in single podocytes. Compared with the findings by high-throughput scRNA-seq, combined nested PCR with Sanger sequencing demonstrated a great advantage.

It is worth noting that IgG4, which was the most predominate IgG subclass in the podocyte line [17] and the deposited IgG under the podocytes in primary membranous nephropathy, were also detected in primary single podocytes. Moreover, chronic kidney patients with MN, IgAN and ischemic nephropathy displayed more classes of Igs and V_H_DJ_H_ rearrangement patterns in single podocytes and higher SHM than did the healthy controls, which was worth being explored in different diseases. The possible explanation is that the complex renal microenvironment in chronic kidney disease stimulates podocytes to produce a variety of Igs and promotes immunoglobulin affinity maturation through high frequency of SHM. Our results are supported by the released scRNA-seq data of the kidney, in which Igs are shown to be in glomerular podocytes from human (GSE109564) [23] and wild type C57BL/6 (GSE111107) [24]. 

In general, activated naïve B cells can switch from low-affinity IgM and IgD to high-affinity IgG, IgE or IgA. Isotype switching involves a replacement of the μ/ δ heavy chain constant regions of the Igγ, ε or α constant regions, whereas the V_H_DJ_H_ patterns remain unchanged. Surprisingly, there were no identical patterns between IgM/IgD and the other three Igs in most single podocytes, suggesting that podocyte-derived V_H_DJ_H_ sequences do not originate from the classical class-switch rearrangement. Coincidentally, Shi [19] reported that each Ig class showed unique V_H_DJ_H_ patterns in a single B-cell expressing multiple Ig classes, which strongly supported our results.

In this study, the existence of multiple V_H_DJ_H_ patterns or IgH classes in a single podocyte occurred in some podocytes, especially in podocytes from kidney disease. It should be noted that our findings are mainly at the mRNA level, and whether these findings can be reproduced at the protein level remains unclear. We have hypothesized the following possible mechanisms. First, our previous study demonstrated that recombination activating gene RAG1/2 were detected in human podocyte cell line. We speculated that transposon mechanism, which can be mediated by RAG1, may exist in podocytes, causing multiple V_H_DJ_H_ recombination on an IgH locus. Second, incomplete DNA deletion may occur in some podocytes, resulting in partial removal of the Ig gene segments between the selected V and D segments or D and J segments, which bring about multiple recombination on an IgH locus. Nevertheless, the underlying mechanisms for the phenomenon that more than one V_H_DJ_H_ pattern or IgH class in a single podocyte require in-depth investigation at the genome level.

This is the first time, by combined nested PCR with Sanger sequencing, demonstrating that human podocytes are able to produce Igs in normal conditions and alter the expression in pathological circumstances. Our study has some shortcomings. First, the kidney biopsy consisted of a very small sample of about 10 × 2 mm, among which few podocytes are difficult to use for high-throughput scRNA-seq or flow sorting. Therefore, we manually picked glomerular single cells to amplify IgH and identified them by podocyte marker genes. It is difficult to get enough podocytes to compare with a large number of B cells. In this study, the number of B cells and podocytes is not commensurable (793 vs 48). The numbers of functional V_H_DJ_H_ sequences used for feature analysis are 798 vs 411. It may affect the results to some extent. However, previous studies on non-B Igs have shown that non-B Igs presented limited diversity, including biased VH usage, identical V_H_DJ_H_ sequences in different cells and non-Gaussian distribution. These findings further supported our results. Second, it was hard to detect the protein expression of Igs in single podocytes. The findings that the human podocyte line can produce and secrete IgG protein may compensate for the shortcomings [17]. In summary, our data suggest that a single podocyte is capable of expressing Igs and that podocytes from patients have more co-expression of Igs, more V_H_DJ_H_ rearrangement patterns and higher frequency of SHM. The potential pathological and clinical significance of Igs by podocytes calls for further investigation.

## 5. Conclusions

In our work, we confirmed the transcripts and V_H_DJ_H_ rearrangements of immunoglobulins in primary podocytes from different individuals. Here, we showed that there were five types of immunoglobulin heavy chains in podocytes. Podocyte-derived Igs presented functional V_H_DJ_H_ rearrangements and limited diversity. More importantly, podocytes from patients had more co-expression of Igs, more V_H_DJ_H_ rearrangement patterns and higher frequency of SHM.

## Figures and Tables

**Figure 1 genes-12-00472-f001:**
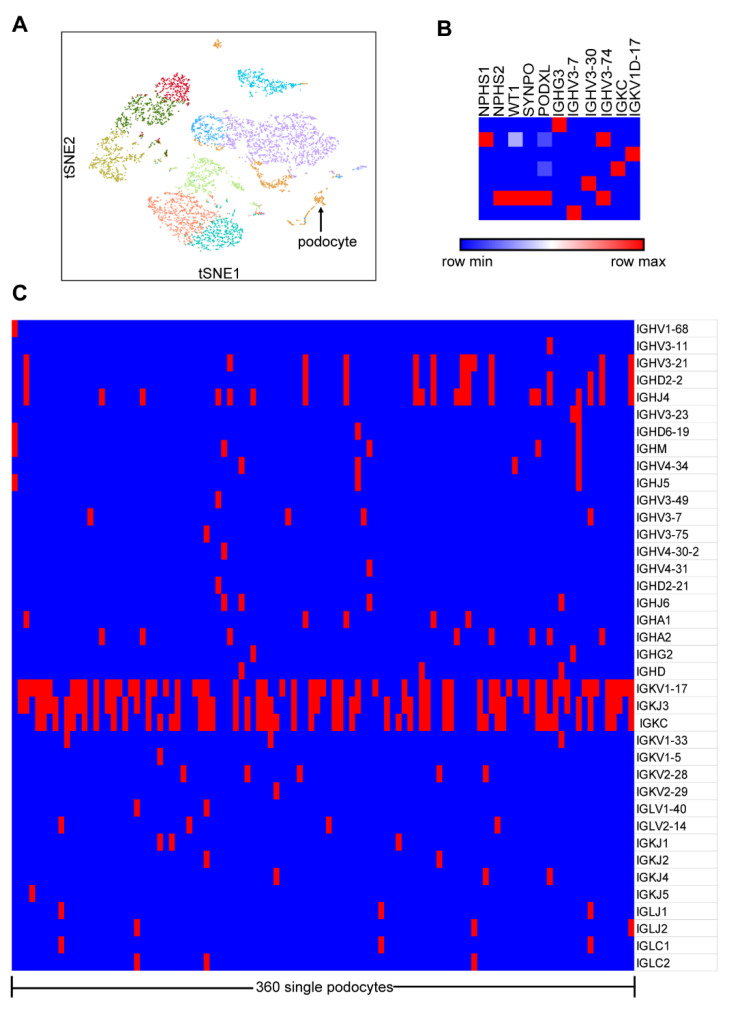
Ig gene transcripts and V, D or J segments were detected in podocytes by 10×Genomics Chromium system. (**A**) The plot shows a two-dimensional representation (tSNE: t-distributed stochastic neighbor embedding) of global relationships of all the sequenced single cells. (**B**) Transcripts of both Ig heavy chains and light chains, constant regions and variable regions were detected in only 7/360 podocytes. (**C**) Transcripts of V, D and J gene segments were detected in 29.4% (106/360) of podocytes. Blue, no immunoglobulins (Igs) in podocytes; red, some Ig gene segments in podocytes.

**Figure 2 genes-12-00472-f002:**
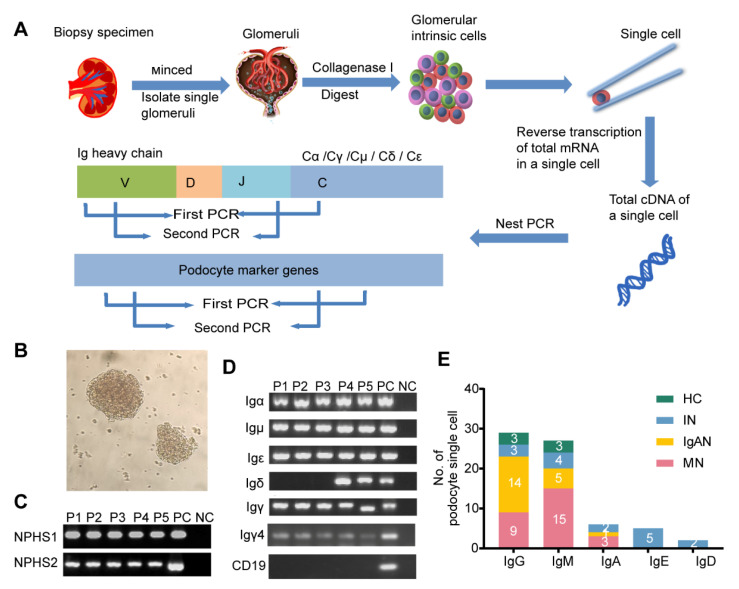
Transcripts of five classes of Ig heavy chain genes in single podocytes. (**A**) Schematic overview of the single-cell sequencing workflow in glomeruli. (**B**) Glomeruli were isolated from the finely minced biopsy specimen. (**C**) Nest PCR analysis of the podocyte marker genes NPHS1and NPHS2. cDNA from kidney cortex used as a positive control (PC); water instead of cDNA was used as a negative control (NC). P, podocyte. (**D**) Five classes of Ig heavy chain genes were detected in podocytes. Peripheral blood mononuclear cells (PBMCs) were used as a positive control (PC); water instead of cDNA was used as a negative control (NC). A CD19 transcript was not detected in any of the podocyte single-cell cDNA libraries, but was detected in PBMCs. (**E**) In the 48 single podocytes, 29 were IgG positive, 27 were IgM positive, 6 were IgA positive, 5 were IgE positive and 2 were IgD positive. The different groups are marked by the bands. HC, healthy control; IN, ischemic nephropathy.

**Figure 3 genes-12-00472-f003:**
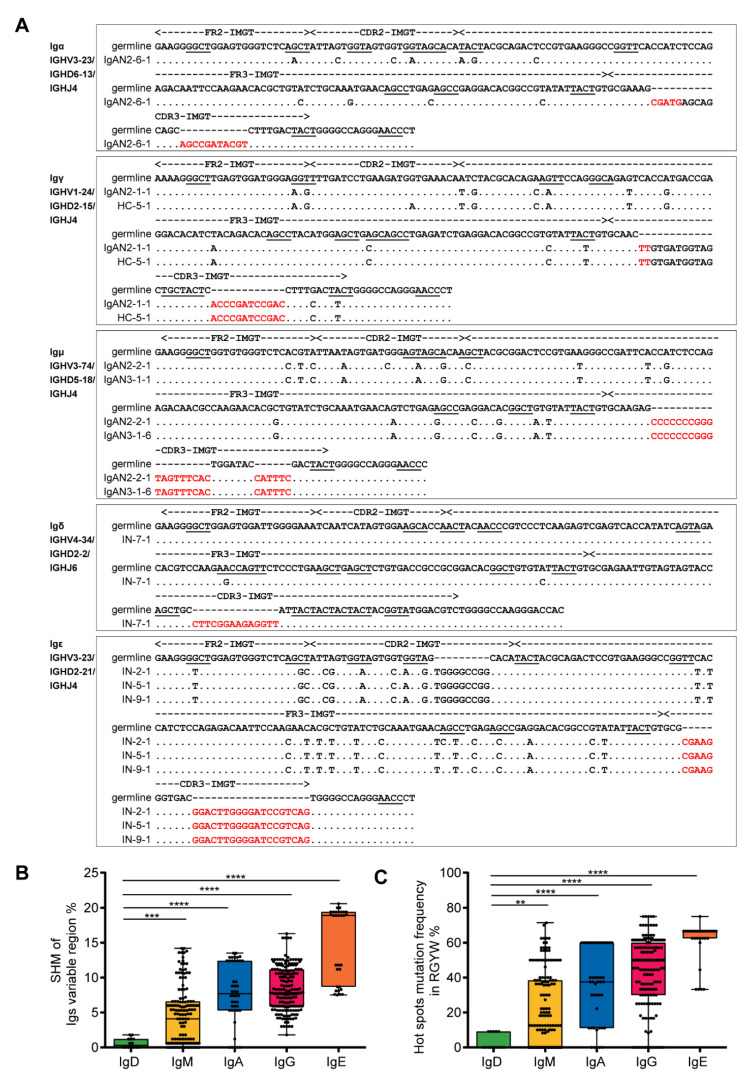
V_H_DJ_H_ rearrangements and somatic hypermutation (SHM) in single podocytes similar to those features in B cells. (**A**). Examples of Ig variable region sequences and mutations in Igα, Igγ, Igμ, Igδ and Igε were illustrated. The sequences of the V_H_DJ_H_ rearrangement were aligned and compared with the most homologous germline sequences, as indicated by dots. Each nucleotide mutation is indicated. The red letters refer to the junctions. The hot spots of mutation in germline genes are underlined. It is worth noting that some different single podocytes have the same V_H_DJ_H_ rearrangement, such as IgAN2-1 and HC-5 in Igγ; IgAN2-2 and IgAN3-1 in Igμ; and IN-2, IN-5 and IN-9 in Igε. HC, healthy control; CDR, complementarity determining region; FR, framework region. Somatic hypermutation (**B**) and hotspot motif mutation (**C**) of the variable regions in 411 V_H_DJ_H_ sequences from 48 single podocytes were evaluated. IgD, n = 14; IgM, n = 150; IgA, n = 44; IgG, n = 172; IgE, n = 31. ** *p* < 0.001, *** *p* < 0.0001, **** *p* < 0.00001.

**Figure 4 genes-12-00472-f004:**
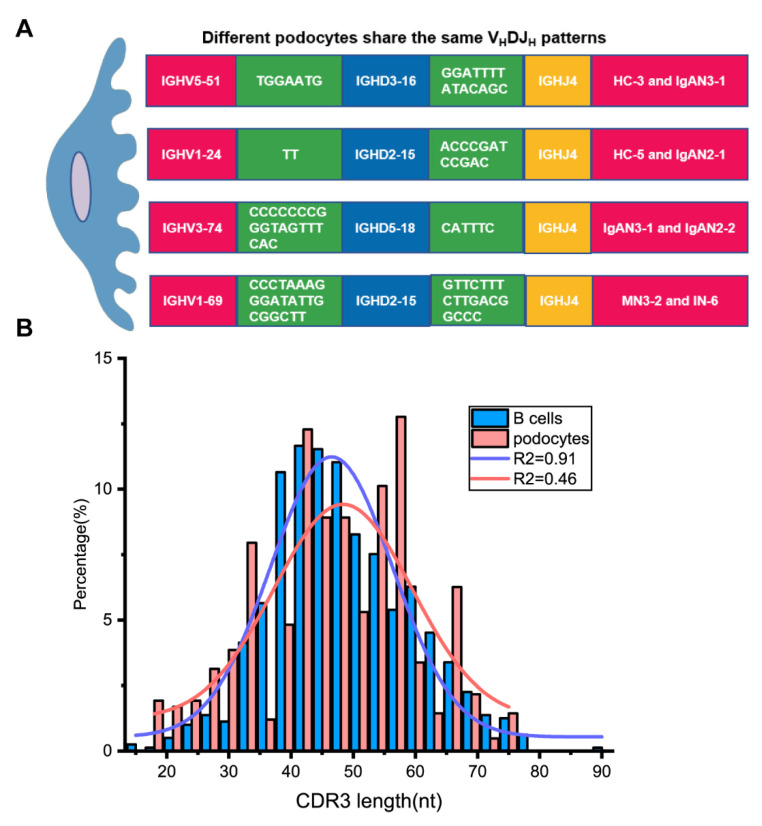
Podocyte-derived Igs displayed less diversity than B-Igs. (**A**). Four pairs of single podocytes shared the same V_H_DJ_H_ pattern in different individuals. The nucleotides in the green rectangle on both sides of the D region are the incorporated N bases. (**B**) The CDR3 length distribution of podocytes differs from that of B cells. The distribution of CDR3 length in B cells fits normal distribution (*R*^2^ = 0.91) while the CDR3 length in podocytes displayed non-normal distribution (*R*^2^ = 0.46). nt = nucleotide.

**Figure 5 genes-12-00472-f005:**
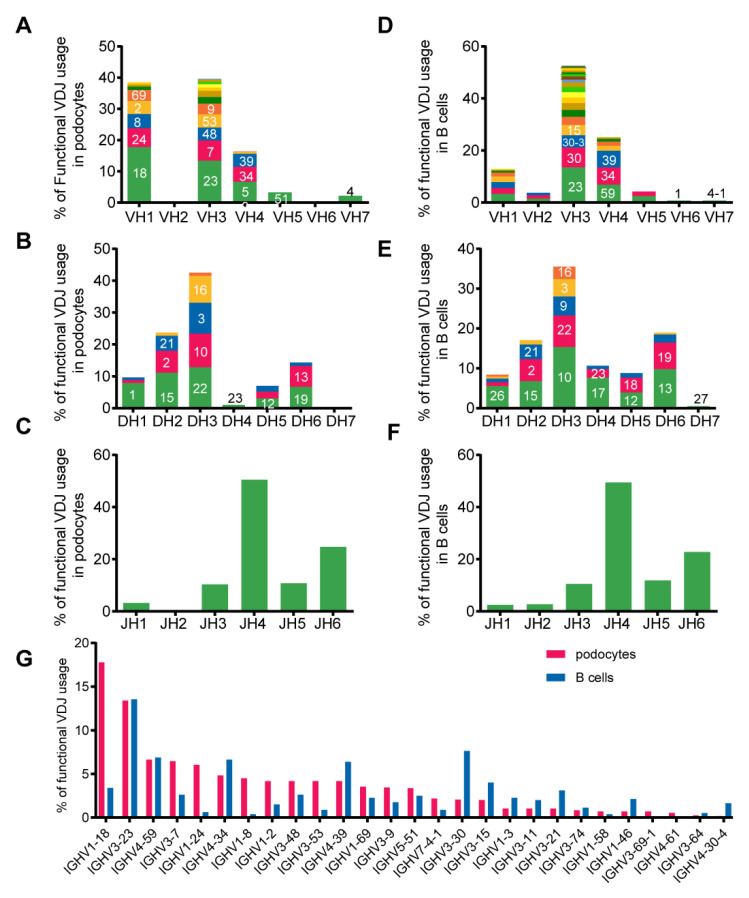
Podocyte-derived Igs displayed biased VH1 gene family usage in single podocytes. We analyzed the usage of functional V genes from the podocyte-derived Igs and B-Igs. The histograms show the relative percentages of total sequences for each of the IgH variable region families for the podocytes and B cells. Within each family, discrete bands represent each of the individual genes. The most abundant genes within each family are indicated (e.g.,18 in VH1 refers to the gene IGHV1-18). (**A**–**C**) The VH, DH and JH gene family usage data from 48 podocytes. (**D**–**F**), The VH, DH and JH gene family usage data from B cells. (**G**)The frequencies of VH gene segments used by podocytes were different from those of B cells.

**Figure 6 genes-12-00472-f006:**
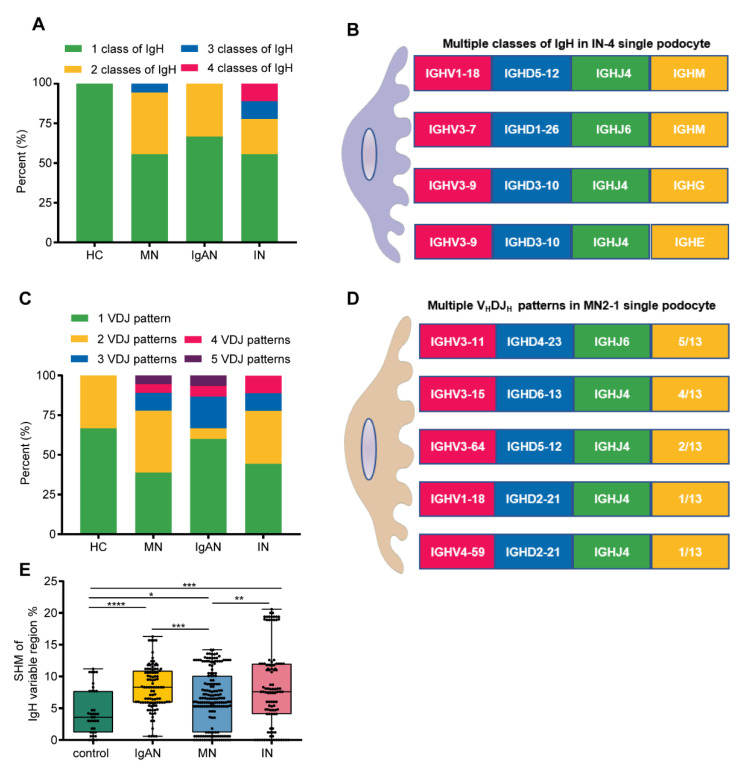
Podocytes from kidney patients exhibited more Ig classes, V_H_DJ_H_ patterns and SHM. (**A**,**B**). Proportions of podocytes expressing Ig classes from healthy control, MN, IgAN and IN subjects. Podocytes from kidney patients exhibited multiple Ig classes in a single podocyte; for example, IN-4 podocyte displayed 3 classes of Ig H. (**C**,**D**). Proportions of podocytes expressing V_H_DJ_H_ rearrangement patterns from healthy control, MN, IgAN and IN. Podocytes from kidney patients exhibited multiple V_H_DJ_H_ patterns in a single podocyte; for example, MN2-1 podocyte displayed five V_H_DJ_H_ rearrangement patterns (**E**). Somatic hypermutation frequency of functional V_H_DJ_H_ sequences in kidney patients was higher than that in the healthy controls. Control, n = 37; IgAN, n = 99; MN, n = 174; IN, n = 101.Significance was determined by Mann-Whitney U test. * *p* < 0.05, ** *p* < 0.01, *** *p* < 0.001, **** *p* < 0.0001.

**Figure 7 genes-12-00472-f007:**
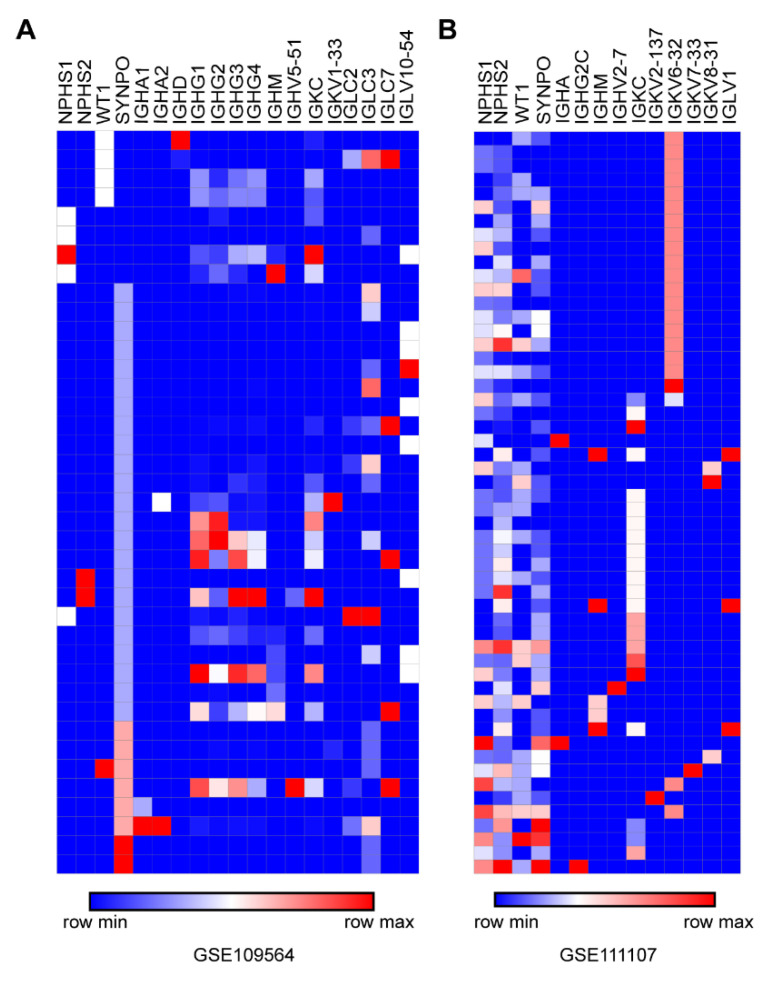
Confirmation of Ig transcription in podocytes by published single-cell sequencing datasets for both humans and mice. (**A**). Ig transcripts in podocytes were found from the scRNA-seq database of an allograft kidney biopsy undergoing mixed rejection (GSE109564). (**B**). Ig transcripts in podocytes were also found from the scRNA-seq database of isolated glomerular cells from C57BL/6 (GSE111107).

## Data Availability

The datasets used and/or analyzed during the current study are available from the corresponding author on reasonable request.

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
