# Peer review of "Single-Cell Sequencing Confirms Transcripts and V_H_DJ_H_ Rearrangements of Immunoglobulin Genes in Human Podocytes"

_genes, 2021, doi:10.3390/genes12040472_

Round 1

Reviewer 1 Report

The authors present data on transcripts and VHDJH rearrangements of immunoglobulin genes in podocytes. They used single cell RNA sequencing by the 10xGenomics Chromium system of cell suspensions from normal kidney cortex adjacent to renal cancer and detected Ig transcripts in 7/360 podocytes and Ig gene segments in 106/360 podocytes. They then combined nested PCR with Sanger sequencing to detect the transcripts and characterize the repertoires of Igs in 48 single podocytes and found that five classes of Ig heavy chains were amplified in podocytes. 429 VHDJH rearrangement sequences were analyzed and showed that podocyte-derived Igs exhibited classic VHDJH rearrangements with nucleotide additions and somatic hypermutations, biased VH1 usage and restricted diversity. Compared with the podocytes from healthy controls that usually expressed one class of Ig and one VHDJH pattern, podocytes from patients with kidney disease expressed more multiple classes of Ig, VHDJH patterns and higher somatic hypermutation. The authors suggest that podocytes express Ig in normal conditions and increase Ig diversity in pathological situations.

The authors present interesting and robust data that is supported by existing scRNA-seq datasets in which Igs were detected in glomerular podocytes from human and wild type C57BL/6 mice. The number of individuals for each condition was small, but as a proof-of-principle study is acceptable. It would be useful to discuss more why the nested PCR approach was superior to the 10xGenomics platform in this setting.

Author Response

Thanks for your suggestion, we added the following words to the discussion of the manuscript and marked in red, as shown in Line 321-331.

“To be specific, the IgG gene abundance of non-B cells is relatively low, so the 10× Genomics technology is not sensitive enough to the detection of low abundance genes. Therefore, the evidence for Igs gene transcription was obtained from such a small number of podocytes although high throughput sequencing technologies were utilized in this study and in other researches on scRNA-seq of kidney. Nested PCR amplified by two rounds of PCR increased the sensitivity of detecting low abundance genes. In addition, because of the diversity of immunoglobulin gene rearrangements, it is difficult to re-assemble the intact V(D)J sequences which has been disrupted into small segments before sequencing. Sanger sequencing directly sequenced large fragments of Ig genes, without interruption in advance, avoiding the problems caused by Ig gene segmentation and reassemble.”

Reviewer 2 Report

The manuscript by Deng et al describes transcriptome based profiling of IG VDJ rearrangements in podocytes at single-cell level using 10x Genomics and more conventional, PCR + Sanger sequencing based approaches. The presented study is valuable, the authors invested subtantial time and effort in this project.

Comments:

  • In Methods 2.1 second paragraph, 2+4+3+1=10 patients are mentioned while Table SI only contains information about 8 patients. How was kidney cortex as healthy control obtained for the study?
  • It would be nice to see matching podocyte vs B-cell data from the same patients. 
  • Results 3.1: '...both heavy chain and light chain as well as constant region and variable region...' I would remove the word 'both' and would change the second 'and' to 'or'. The way it's currently written suggests as if multiple of these regions would have been detected in the same cell(s). 
  • Results 3.4 section titel: 'Podocyte-derived Igs displayed VH1 bias and lower diversity than B-Igs' would be better.
  • Fig 4A: It would be good to see the incorporated N bases.
  • Fig 4B: Were the number of B-cells vs number of podocytes commensurable? If not, this fact may well have affected the conclusion made based on this histogram. (Too high number of categories compared to the number of podocytes analyzed may have caused this non-Gaussian distribution)  
  • Results 3.5: Was really a percentage calculated based on 6 cells?
  • Fig 6D: Based on th IGH gene map, it seems to me that at least three alleles are needed for these 5 rearrangements. Do we know anything about the number of chromosomes 14 in these podocytes?
  • Table SV is informative but it would be useful to see the full CDR3 sequences incl N bases in the supplement.
  • Discussion should include more explanation and interpretation on the findings, e.g. the higher number of Igs and VDJH rearrangements in patients with renal disease.
  • There are some typos in the text, a native speaker with fresh eyes should read and correct the text.

Author Response

Question 1: In Methods 2.1 second paragraph, 2+4+3+1=10 patients are mentioned while Table SI only contains information about 8 patients. How was kidney cortex as healthy control obtained for the study?

Answer: Normal renal cortexes were collected from donors undergoing nephrectomy as a result of renal cell carcinoma. In this study, we obtained normal renal cortexes from 2 patients with renal cell carcinoma, of which one was used for 10x Genomics and the other one was used for nested PCR and Sanger sequencing. Only the patients used for nested PCR and Sanger sequencing was mentioned in Table SI. Thank you for your suggestion. We added this patient's information to the table SI, marked in red.

Question 2: It would be nice to see matching podocyte vs B-cell data from the same patients. 

Answer: We agree with you. We also think it will be better to match podocytes vs B-cell data from the same patients. But it is difficult to obtain enough peripheral blood to sort B cells by FACS for high-throughput sequencing. In addition, obtaining a large amount of peripheral blood from kidney patients for scientific research is difficult to pass ethics committee review.

Question 3: Results 3.1: '...both heavy chain and light chain as well as constant region and variable region...' I would remove the word 'both' and would change the second 'and' to 'or'. The way it's currently written suggests as if multiple of these regions would have been detected in the same cell(s). 

Answer: Thank you for your suggestion. I removed the word 'both' and change the second 'and' to 'or',as shown in Line 161.

Question 4: Results 3.4 section title: 'Podocyte-derived Igs displayed VH1 bias and lower diversity than B-Igs' would be better.

Answer: Thank you for your suggestion. We revised the Results 3.4 section title, as shown in Line 225.

Question 5: Fig 4A: It would be good to see the incorporated N bases.

Answer: Thank you for your suggestion. We added the incorporated N bases in the Fig 4A. The nucleotides in the green rectangle on both sides of the D region are the incorporated N bases. Figure 4A legend has also been modified, as shown in Line 237-238.

Question 6: Fig 4B: Were the number of B-cells vs number of podocytes commensurable? If not, this fact may well have affected the conclusion made based on this histogram. (Too high number of categories compared to the number of podocytes analyzed may have caused this non-Gaussian distribution)  

Answer: It is difficult to get enough podocytes to compare with a large number of B cells Indeed, the number of B-cells and podocytes is not commensurable (793 vs 48). The numbers of functional VDJ sequences used for Gaussian distribution analysis are 798, 411 respectively. It may affect the results to some extent. However, previous studies on non-B Igs have shown that non-B Igs presented limited diversity, including biased VH usage, identical VDJ sequences in different cells and non-Gaussian distribution. These findings further supported our results.

 We added the explanation to the discussion (Line 387-393), marked in red.

Question 7: Results 3.5: Was really a percentage calculated based on 6 cells?

Answer: In this study, we did detect the IgG gene transcription and rearrangement in only six “normal” podocytes. Subsequently, 37 sequences of VHDJH rearrangements from the 6 “normal” podocytes were sequenced and analyzed, and found that only one class of IgH in all the 6 “normal” podocytes. By analyzing 37 VHDJH rearrangement sequences from the 6 “normal” podocytes, we found that 2/6 podocytes expressed more than one functional VHDJH pattern. In addition, the SHM frequency in normal podocytes was significantly lower than those in podocytes from patients with kidney disease. The small number of normal podocytes was indeed a limitation of this study, but it was very difficult to get more podocytes.

Question 8: Fig 6D: Based on the IGH gene map, it seems to me that at least three alleles are needed for these 5 rearrangements. Do we know anything about the number of chromosomes 14 in these podocytes?

Answer: Sorry, we do not know anything about the number of chromosomes 14 in these podocytes. We have speculated and explained the possibility of the phenomenon that more than one VHDJH rearrangements in a single podocyte, marked in red, as shown in Line 368-380.

In this study, the existence of multiple VHDJH patterns or IgH classes in a single podocyte occurred in some podocytes, especially in podocytes from kidney disease. It should be noted that our findings are mainly at the mRNA level, and whether these findings can be reproduced at the protein level remains unclear. We have hypothesized the following possible mechanisms. First, our previous study demonstrated that recombination activating gene RAG1/2 were detected in human podocyte cell line. We speculated that transposon mechanism, which can be mediated by RAG1, may exist in podocytes, causing multiple VHDJH recombination on an IgH locus. Second, incomplete DNA deletion may occur in some podocytes, resulting in partial removal of the Ig gene segments between the selected V and D segments or D and J segments, which bring about multiple recombination on an IgH locus. Nevertheless, the underlying mechanisms for the phenomenon that more than one VHDJH pattern or IgH class in a single podocyte require in-depth investigation at the genome level.

Question 9: Table SV is informative but it would be useful to see the full CDR3 sequences incl N bases in the supplement.

Answer: Thanks for your suggestion. We inserted the CDR3 sequence information including N bases into the Table SV, as shown in revised supplementary material.

Question 10: Discussion should include more explanation and interpretation on the findings, e.g. the higher number of Igs and VDJ rearrangements in patients with renal disease.

Answer: Thanks for your suggestion. We added more explanation and interpretation on the findings to the discussion. We discussed the advantages of nested PCR and Sanger sequencing over 10× Genomics technology, as shown in Line 321-331. We hypothesized the possible mechanisms for the phenomenon that more than one VHDJH pattern or IgH class in a single podocyte, as shown in Line 368-380 In addition, we explained the possible reasons for the result that higher number of Igs and VHDJH rearrangements in patients with renal disease, as shown in Line 354-356.

Question 11: There are some typos in the text, a native speaker with fresh eyes should read and correct the text.

Answer: Thanks for your suggestion. Considering that English is not our native language, we turn to language editing services (MDPI) to improve the language of our article.

Reviewer 3 Report

“Single-cell Sequencing Confirms Transcripts and VHDJH  Rearrangements of Immunoglobulin Genes in Human Podocytes” by Deng et al. Discuss the stack of Igs in primary podocytes cells. The paper is clearly introduced and easy to follow, the introduction section contains the necessary background of the topic and the methods section is clearly described. I have a minor suggestion is to change the box plot point to be smaller for clarity. 

  Deng et al. in their paper “Single-cell Sequencing Confirms Transcripts and VHDJH Rearrangements of Immunoglobulin Genes in Human Podocytes” aimed to characterize repertoires of Igs in primary podocytes at the single-cell level, compared to what already known that it is solely produced by B cells. Additionally, they tried to confirm it by studying transcripts. That lead to the confirmation of the Ig transcript existence in   7/360 podocytes and Ig gene segments in 16 106/360 podocytes. Furthermore, they found that five classes of Ig heavy 18 chains were amplified in podocytes by using PCR amplification and Sanger sequencing of 48 single podocytes. These findings by Deng et al. enrich our knowledge regarding Igs production.   The introduction section is well written and gives a clear background about the topic. The methods section describes carefully each step in detail. The results and discussion is clear and supports the conclusion, a minor suggestion is to change the box plot in the results section to make the dots smaller for clarity.

Author Response

Thanks for your suggestion, we have changed the box plot point to be smaller (from Size 2 to Size 1). At the same time, we have replaced Figure 3 and Figure 6, in which the box plot point became smaller.
